# Understanding the association between parental attitudes and the practice of female genital mutilation among daughters

**Claudia Cappa****\*, Claire Thomson, Colleen Murray**

Data and Analytics Section, Division of Data, Analytics, Planning and Monitoring, United Nations Children's Fund, New York, NY, United States of America

\* ccappa@unicef.org

**Data Availability Statement:** All the underlying datasets used for the analysis can be found on DHS website, at: https://www.dhsprogram.com/

## Abstract

Female genital mutilation is a harmful traditional practice that violates girls' right to health and overall well-being. Most research cites social acceptance, marriageability, community belonging, proof of virginity, curbing promiscuity, hygiene, and religion as motivations for the practice. It is generally assumed that individual attitudes of parents and other family members have an impact on decisions related to the cutting of girls, and that such attitudes are influenced by social norms. The aim of this study is to understand how parental attitudes towards the practice of female genital mutilation influence decision making related to the cutting of girls. Data from 15 Demographic and Health Surveys were analyzed to assess whether couples with at least one living daughter aged 0 to 14 years share the same opinions about the continuation of the practice, and to what extent couples' opinions are associated with the risk of daughters being cut. The analysis reveals that a significant percentage of couples hold discordant opinions on the continuation of the practice including in countries where the practice is very common. While a daughter's likelihood of being cut is much higher when both parents think the practice should continue, the analysis also shows that many cut girls have parents who oppose the practice. It further suggests that female genital mutilation is more prevalent among daughters whose mothers want the practice to continue and whose fathers are opposed or undecided, compared to daughters with fathers who are the sole parent supporting its continuation. Understanding the extent to which parental opinions influence decisions and which girls are most likely to be cut is essential for developing appropriate interventions aimed at promoting the abandonment of the practice.

## Introduction

The World Health Organization defines female genital mutilation (FGM, also referred to as "female circumcision" and "female genital cutting") as "comprising all procedures that involve partial or total removal of the external female genitalia, or other injury to the female genital organs for non-medical reasons" [1]. There are many documented health complications caused by FGM, including pain and trauma during the procedure, risk of infection, potential

**Funding:** The author(s) received no specific funding for this work.

**Competing interests:** The authors have declared that no competing interests exist.

loss of life, and other problems that jeopardize the reproductive health of the woman through-out her life (with particular challenges for both woman and child during pregnancy and child-birth) [2]. Most research cites social acceptance, marriageability, community belonging, proof of virginity, curbing promiscuity, hygiene, religion, and tradition as motivations for the practice [3]. FGM is practiced primarily in Africa, although prevalence varies dramatically between (and often within) countries, from less than 1% in Cameroon and Uganda, to over 90% in Guinea and Somalia [4]. As of 2016, the United Nations Children's Fund estimates that at least 200 million girls have been subjected to the practice in the 30 countries where the practice is most prevalent and for which representative data are available [5].

Despite its centuries-long existence, FGM is becoming less common [5,6]. Many measures by governments and organizations aimed at fostering abandonment of the practice have con-tributed to its decline. A review of FGM interventions by Johansen, Diop, Laverack, and Leye found that the seven most common approaches were (1) health risk education approaches, (2) conversion of circumcisers, (3) training of health-care professionals as agents of change, (4) the creation of alternative rituals, (5) community-led approaches, (6) public statements, and (7) legal methods [7]. The authors found that the interventions that had the greatest successes were those that used several methods in conjunction. They argue that the local context and source of decision-making power must be considered for an intervention to be successful [7].

The need for multi-pronged interventions is echoed in a multitude of other literature. The overlay of intervention techniques was highlighted by Shell-Duncan, Hernlund, Wander, and Moreau through their research in communities in Senegal, highlighting how a federal law criminalizing FGM did not in itself contribute to a reduction in prevalence. It was only when community-based interventions were run that headway in curbing prevalence was made [8]. This finding is supported by Camilotti's research, which found that passage of the law crimi-nalizing FGM in Senegal correlated to a decrease in the age at which girls were circumcised. She attributed this change to a lack of programs or interventions addressing the social causes of FGM, causing the law to merely force the practice underground [9]. Additionally, Crisman, Dykstra, Kenny and O'Donnell found that the implementation of a law criminalizing FGM in Burkina Faso was effective at helping reduce cases of FGM, but stressed this success was due to concentrated, long-term advocacy and enforcement surrounding the law [10]. Similar multi-pronged interventions have been successful in Sudan [11].

In all of the above interventions, recognition of local context and community attitudes has been central to their success. It is generally assumed that individual attitudes have an impact on decisions related to the cutting of girls, and that such attitudes are influenced by social norms, hence the focus on interventions aimed at addressing attitudes to promote abandon-ment of FGM. The available literature suggests that, in most contexts, mothers and other female family members are responsible for the circumcision of girls; however, fathers and other male family members are becoming increasingly relevant, and in some cases play a dom-inant role, both in favor of and against the practice.

Bjälkande, Bailah, Harman, Bergström, and Almroth found that in Sierra Leone, mothers and fathers were mentioned equally as decision makers for FGM, and that the role of male family members was on the rise [12]. In Jigjiga Town, Ethiopia, researchers found that the mother was the primary decision maker in 67% of FGM cases, with the decision being made jointly between parents 24% of the time [13]. A survey at Khartoum University in Sudan revealed that 21% of women who had been cut reported that it was their father's decision, com-pared to 65% who said it was predominantly their mother's choice [14]. In Hargeisa, Somali-land, Lunde and Sagbakkeb provided evidence that mothers who had daughters in more recent years were more susceptible to the opinions of husbands and fathers compared to moth-ers of previous generations, and called for the recognition of gender relations in changing the

practice [15]. Research in Kano, in Northern Nigeria, found that 84% of mothers did not support the continuation of FGM, but that it continued as fathers were the main decision makers about their daughters' circumcision, and they maintained high support for the practice [16]. On the other hand, Abusharaf recounted a case in Sudan of a man who pressed criminal charges against his wife for circumcising their two daughters when he was out of the country [17]. Hernlund and Shell-Duncan found that in Senegal and the Gambia, when fathers were involved in decision making around FGM, their daughters were more likely to remain uncut [18].

Insights into the ways in which couples' opinions impact decisions and behaviors related to FGM provide an important avenue to design more effective interventions. FGM exists as a social norm, however, and the decisions of individuals must be understood within that wider framework of collective influences. Mackie stated that FGM will continue for as long as parents perceive that the harm of cutting their daughter is outweighed by the consequences of not cutting her [19]. To not be cut, and thus risk social ostracism and exclusion from benefits (such as the marriage market), is too great a risk for parents to take with their child; thus, they will perpetuate the practice. He argued that the costs of one person breaking the norm are too high for an individual to attempt, and that the end of a norm requires group consensus, or at least group action, to ensure that a norm can be abandoned without fear of significant backlash [19]. Hernlund and Shell-Duncan disputed Mackie's approach by arguing that it is not as simple as a "for" or "against" FGM argument, but that people have significant flexibility and fluidity in their choices, which are shaped by a multitude of factors and cannot be dichotomized. They argued that families are constantly evaluating their beliefs as they try to position themselves within constantly shifting relationships, experiences, and information [18].

Cloward built on the multifaceted approach proposed by Hernlund and Shell-Duncan by examining the ways in which individuals are influenced by social norms with regards to change [20]. Cloward argued that an individual goes through a balance shifting of norms, from agreeing to opposing, based on the factors surrounding that individual. She contended that individuals better able to mitigate the cost of defection from the norm will be the fastest to abandon the norm, therefore lessening the social cost of defection for those less able to mitigate the costs, and that norm change is an incremental process, as opposed to the "critical mass" favored by Mackie [19].

By building on the approaches proposed by Mackie, Hernlund and Shell-Duncan, and Cloward, there is space to adapt their theories into a microscale analysis of a couple contemplating FGM as a social norm. Cloward argued that those better able to mitigate the social costs of breaking a norm will likely be the first to defect [20]. Following this thread, it could be that fathers will be more willing to break with the social convention of cutting daughters. They are less familiar with the social fallout and will be less likely to be held accountable for the backlash of not being cut. While fathers might be sheltered from societal backlash for not cutting, however, they may not necessarily be shielded from the backlash of a disagreeing partner. Conversely, a mother opposed to cutting her daughter might be overruled by a father who supports the practice and is enabled as the final decision maker. Alternatively, in a situation where both partners are in concurrence, the strength of the societal norm (either for or against FGM) could still overrule their individual preferences, placing them in Hernlund and Shell-Duncan's categories of being "reluctant adherents" or "reluctant abandoners" of the practice [18].

In situations where there is discordance of opinions within couples, there is a potential for negotiation. This negotiation could take many forms, from one partner attempting to dissuade the other for the sake of preventing the cutting, to arguing for a delay in the age at cutting, a less severe form of cutting or having the procedure done in what is perceived to be a safer environment. Dunbar and Burgoon argued that this type of negotiation will rest heavily on the existing power dynamic, which is created by the relationship itself as well as the norms of the

culture in which it exists [21]. They argued that this power dynamic is crucial, as in many situations it will inform whether the less powerful partner will address a conflict. Often, in cases of a significant imbalance, partners do a cost-benefit analysis and choose to avoid a minor conflict that might lead to greater conflict in the relationship [21].

When looking at couples' preferences and social norms, and their influence on FGM, a two-fold framework connects opinions to behaviour. First, individuals must form an opinion regarding the cutting of their daughters, weighing the potential costs and benefits. Second, they decide to either cut or not cut their daughters, depending on their agreement, negotiation, a unilateral decision, or overarching societal norms. In this process, individuals may communicate their belief to their partner and, in the event of discordance, decide to enter a state of negotiation. It also is possible, however, that individuals in couples do not communicate their beliefs to one another, or do not discuss their decision, which happens when one partner is responsible for arranging the cutting, and either doesn't consult the other or assumes his/her assent. Within this framework norms are in constant flux, and individuals' opinions are continually shaped by their perceptions of society, and the potential costs or benefits of their actions. As a result, patterns cannot necessarily be generalized within couples as different relationships have different power dynamics, but at the population level, it could be possible to gain insights on how social norms influence decision making around FGM.

With this objective in mind, this study analyzes available survey data to establish whether there is a relationship between the opinions of couples regarding FGM and the actual cutting of their daughters. Building on earlier analyses [4], the paper first looks at the extent to which couples share similar opinions on the practice, and then examines the risks of being cut among daughters depending on parental opinions. Understanding the extent to which parental opinions influence decisions and which girls are most likely to be cut is essential for developing appropriate interventions aimed at promoting abandonment of the practice.

## Materials and methods

This study used data from Demographic and Health Surveys (DHS). The standard DHS questionnaire collects information about the experience of FGM among female respondents aged 15 to 49 years and their daughters aged 0 to 14 years. Nearly all surveys also ask respondents at what age they underwent FGM, by whom, and what specifically was done to them. FGM takes many different forms in many countries, but three main types are covered by a DHS. Respondents are asked whether their genital parts were nicked only, if any flesh was removed, and if the vaginal opening was sewn closed. Mothers are then asked the same questions about their daughters' experiences, covering age, practitioners, and type. Most surveys include additional questions on attitudes asked of all women and men aged 15 to 49 who have heard about the practice. Respondents are asked whether they think that the practice should continue or stop. Of the 25 countries that have collected data on FGM as part of DHS surveys, information on opinions about the practice from both men and women was available for 16 countries. Access to the dataset was restricted for one country, leaving 15 countries which are featured in this analysis.

The first level of analysis compared the opinions of men and women who were a couple and had at least one living daughter aged 0 to 14. Opinions were recoded into new variables to allow for distribution of couples, by whether both agreed that FGM should continue, both agreed that FGM should end, both were undecided, or they had discordant opinions. The discordant category was further recoded into four additional variables. The first two differentiated whether it was the woman or the man who wanted FGM to continue while their partner held a different opinion (i.e., wanted FGM to stop or was unsure), and the second two

indicated whether it was the woman or the man who wanted FGM to stop, while their partner was undecided.

The second level of analysis looked at the association between parents' opinions and the cutting of their daughters, using a subset of the attitudinal categories mentioned above. To determine the occurrence of FGM among daughters, all three types of cutting covered in the questionnaire were combined and treated as a single category in the analysis.

Prevalence estimates among women aged 15 to 49 were used to divide the countries into three categories (Table 1): low-prevalence countries (with a prevalence of 15% or below), mid-prevalence countries (with a prevalence between 16% and 50%), and high-prevalence countries (with a prevalence above 50%). Countries were ranked by prevalence level within each category, from lowest to highest. Prevalence among women aged 15 to 49 was preferred over prevalence among girls aged 0 to 14 when the objective was to give a sense of how widespread the practice is and for ranking countries. The full extent of FGM cannot be captured by looking at the prevalence among daughters aged 0 to 14, given that a daughter in that age range may not be cut simply because she has not yet reached the customary age at cutting.

Analysis was weighted to account for the sampling design of each survey. The p-values presented are for a two-tailed test, with the null hypothesis being that the prevalence of FGM among daughters does not depend on parents' opinions. Confidence intervals were set at 95%.

Estimates based on fewer than 25 unweighted cases were suppressed while those based on 25 to 49 unweighted cases are presented between parentheses. SPSS software version 25 was used to conduct the analysis. Datasets were accessed through the Demographic and Health Survey Program website at https://dhsprogram.com/.

**Table 1. Data sources and categorization of countries by prevalence levels.**

| | Source | Prevalence among girls and women aged 15 to 49 years | Prevalence among girls aged 0 to 14 years[*] |
|---|---|---|---|
| **Low-prevalence countries** | | | |
| Cameroon | DHS 2004 | 1.4 (0.7–2.2) | 0.8 (0.4–1.2) |
| Niger | DHS 2012 | 2.0 (1.3–2.6) | 1.8 (1.2–2.4) |
| Togo | DHS 2013–2014 | 4.7 (3.7–5.6) | 0.3 (0.2–0.5) |
| Benin | DHS 2011–2012 | 7.3 (6.6–7.9) | 0.2 (0.1–0.3) |
| United Republic of Tanzania | DHS 2004–2005 | 14.6 (13.2–16.1) | 4.2 (3.4–5.0) |
| **Mid-prevalence countries** | | | |
| Kenya | DHS 2014 | 21.0 (19.7–22.2) | 2.8 (2.4–3.3) |
| Senegal | DHS 2017 | 24.0 (22.1–25.9) | 13.9 (12.2–15.7) |
| Nigeria | DHS 2013 | 24.8 (23.3–26.3) | 16.9 (15.4–18.4) |
| Côte d'Ivoire | DHS 2011–2012 | 38.2 (35.2–41.1) | 9.8 (8.3–11.3) |
| Chad | DHS 2004 | 44.9 (39.3–50.6) | 20.7 (17.6–23.8) |
| **High-prevalence countries** | | | |
| Ethiopia | DHS 2016 | 65.2 (61.9–68.5) | 15.7 (13.4–18.1) |
| Burkina Faso | DHS 2010 | 75.8 (74.1–77.5) | 13.3 (12.0–14.6) |
| Sierra Leone | DHS 2013 | 89.6 (88.3–90.8) | 31.3 (29.9–32.7) |
| Mali | DHS 2012–2013 | 91.4 (90.2–92.7) | 67.6 (65.4–69.8) |
| Guinea | DHS 2018 | 94.5 (93.6–95.4) | 39.1 (37.3–40.8) |

[*]For Cameroon, Chad, Niger, Sierra Leone and Tanzania, the data refer to the percentage of girls and women aged 15 to 49 years with at least one living daughter who has undergone FGM. Data for Mali from the DHS 2012–2013 are not nationally representative, as some regions were excluded from data collection due to insecurity.

## Results

Fig 1 presents the breakdown of couples by whether they agree or disagree on the continuation of the practice. While most couples report the same opinion on continuation, across all countries with available data, there is great variation in the actual percentages, from 50% of couples sharing the same opinion on FGM in Nigeria and Sierra Leone to close to 90% in Togo. It is interesting to note that significant variation in the proportion of couples who share the same opinion can be found across countries within each of the three prevalence groupings. Among countries with low and medium prevalence, agreement that FGM should stop is the most popular category, except in Chad, where more couples report being in favour of continuation. Most couples in Benin, Burkina Faso, Cameroon, Côte d'Ivoire, Ethiopia, Kenya, Niger, Senegal, Tanzania, and Togo agree on wanting FGM to end, while many couples in Chad, Nigeria, and Sierra Leone have discordant opinions. There was an extremely low proportion of partners who are both undecided about FGM. Among discordant couples, there is an equal split between whether the woman or the man wanted FGM to continue. Noteworthy differences in opinions can be seen in Sierra Leone, where women are significantly more likely to support the continuation of the practice compared to their male counterparts.

Fig 2 shows the prevalence of FGM among daughters by parental opinions, using six of the eight categories of opinion presented in Fig 1. Results are only displayed for countries with moderate and high FGM prevalence due to the small number of cut girls in low-prevalence countries. Similarly, results for the category "both undecided" and "other" are not presented due to the small number of cases falling into these two categories. As expected, in all countries, the prevalence of FGM among daughters has a strong statistical association with parents'

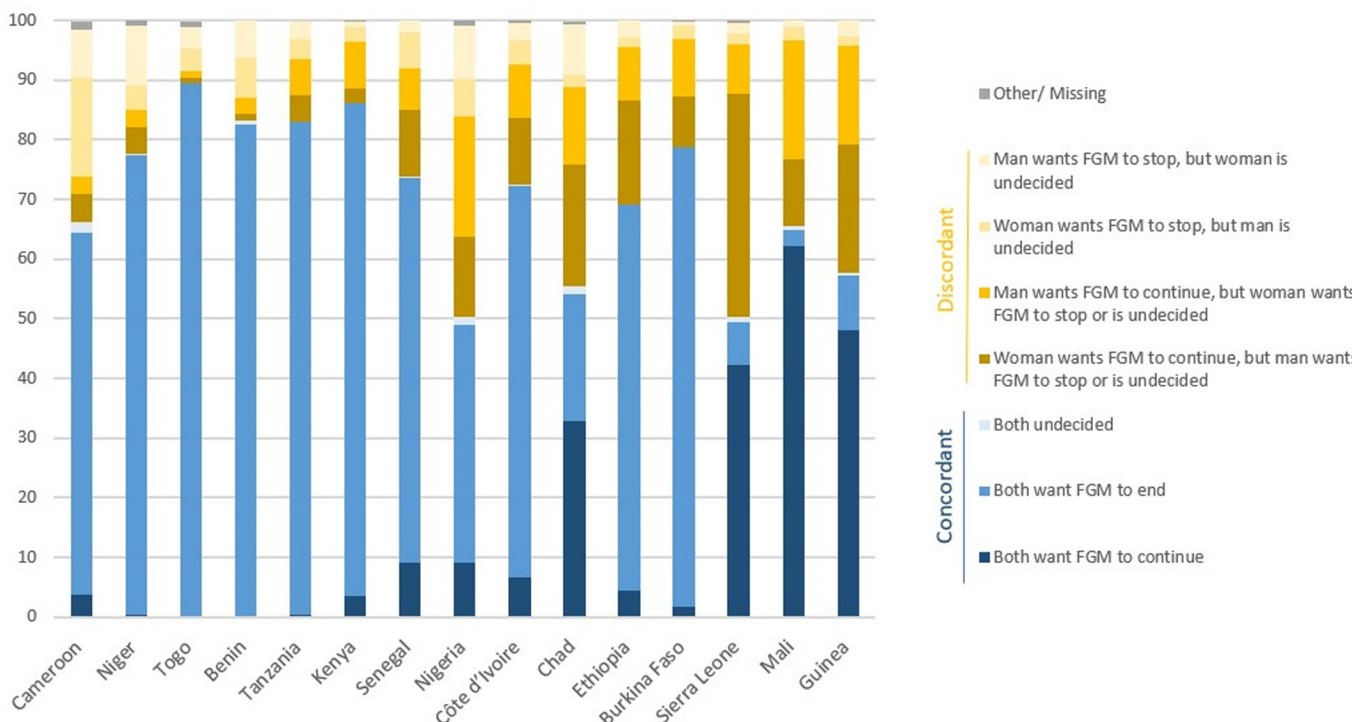

**Fig 1. Percentage distribution of couples with at least one living daughter aged 0 to 14, by whether they have concordant or discordant opinions about the continuation of the practice.** The "other" category includes couples for whom the opinion of one partner is missing, while the "missing" category includes couples for whom the opinions of both partners are missing. More detailed results are in S1 Table.

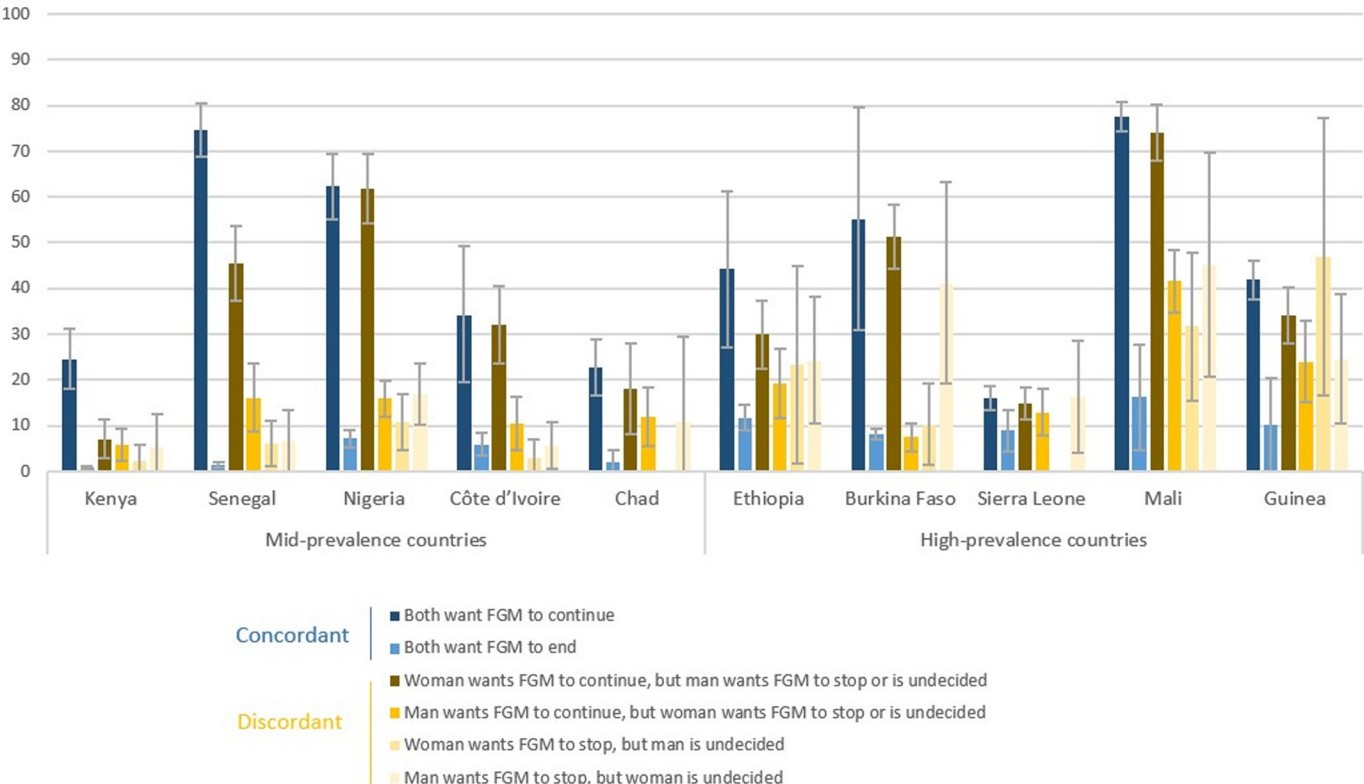

**Fig 2. Percentage of daughters who have undergone FGM, by parental opinions about the continuation of the practice.** More detailed results are included in S2 Table, including p values (<0.0001 for all countries). Missing bars represent results that were suppressed because they were based on fewer than 25 unweighted cases.

opinions. The largest percentage of girls who have undergone the practice is found among daughters of parents who both want FGM to continue, although in many countries (Burkina Faso, Côte d'Ivoire, Guinea, Mali, and Nigeria) this percentage is very close to the one found among discordant couples in which only one parent is supportive. Interestingly, girls are more likely to be cut when the mother wants FGM to continue, even if the father is opposed or undecided, compared to girls whose father want the practice to continue while the mother is opposed or uncertain. Surprisingly, the analysis also shows that many cut girls have parents who are both opposed to the practice. In Mali, for instance, 16% of girls whose parents both want the practice to end were nonetheless cut. Prevalence estimates of cutting among daughters are lowest in cases where either both parents oppose the practice.

## Discussion

These findings support the idea that personal attitudes and social norms play an important role in the determination of whether a girl will undergo FGM, and provide the additional nuance of isolating the impact of opposition of one or both parents. The findings thus reinforce the importance of changing opinions as a step towards eliminating the practice.

The initial analysis of couples' opinions illustrates that concordance of opinion within a household should not be assumed. Further research may investigate the extent to which individuals are aware of their partner's opinions, and in the case of discordance, whether couples actively discuss their attitudes towards FGM. Such research could inform programming

designed to enable community dialogue, with acknowledgment that diversity of opinions may be found even within the same household.

The finding that a girl's risk of FGM decreases with at least one parent who is not in favor of the practice, and further decreases when both are not in favor, underscores the importance of an inclusive approach to social change. The finding that the mother's opinion carries more weight in the determination to perform FGM suggests that efforts to change mothers' opinions may be fruitful in reducing FGM prevalence. Fathers should not be neglected, however, particularly given the gains that could be expected by bringing couples into agreement in their opposition.

Particularly in high-prevalence countries, the sizable proportion of girls undergoing FGM despite their parents being united in opposition to the practice testifies to the strength of societal pressure to conform to the social norm of cutting. While parental opposition to the practice is a protective factor, as these girls are less likely to be cut than their peers whose parents support FGM, opposition alone is not sufficient to prevent all instances of FGM. Further research should seek to identify additional protective factors in high-prevalence contexts, or characteristics of families that do not practice FGM. Such families may resemble those theorized by Cloward as best able to mitigate the costs of defecting from the norm [20].

## Limitations

A variety of limitations needs to be kept in mind when interpreting the findings. Firstly, the available data capture opinions at the time of the surveys, and the analysis assumes that those surveyed hold the same opinion as they did before their daughter was circumcised. This may not necessarily be true, given that it is possible that a couple will have changed opinion based on the experience of their daughters' circumcision, or as a result of other mitigating factors such as awareness campaigns or shifts in community norms.

Another limitation is the reliability of self-reporting by mothers regarding their daughters' circumcision. There is a risk that a woman did not report honestly whether her daughter was cut, particularly in countries where the practice is illegal, out of a desire to keep the cutting concealed. Similarly, both women and men may not provide an honest response about their beliefs. This could be especially true in areas where there have been campaigns against FGM, or where there are social or legal repercussions to having practiced FGM. Finally, this study relies entirely upon parents reporting "for" or "against" the continuation of the practice; however, their opinions cannot be easily dichotomized, as parents may have flexibility and fluidity in their choices, as theorized by Hernlund and Shell-Duncan [18].

Regarding data availability, a limited number of countries could be included in the analysis. Data were only available for 15 countries, with several countries' most recent data being from over a decade ago. While these are the most recent data for each country at the time of the analysis, they cannot be considered reflective of current patterns due to the significant time lapse. Lastly, since the aim of the study was to evaluate the impact of couples' opinions, girls who live with only one parent or with neither parent were not included in the analysis.

## Conclusions

This study is the largest compilation of statistical data on parental opinions towards FGM and their association with the cutting of daughters. The findings indicate that there is a significant percentage of couples with discordant views on whether the practice should continue, including in countries where the practice is very common, which refutes the assumption of concordance of opinions among partners.

The analysis also confirms the association between daughters' cutting and favorable parental opinions towards the practice. In most countries, the highest prevalence of FGM is among daughters of couples who both wanted FGM to continue. High levels of prevalence among discordant couples is a significant finding, however, as it can serve as an insight into who is the key decision maker in a relationship when it comes to daughters' circumcision, and the effect that this decision maker might have upon the outcome. The findings regarding prevalence of cutting suggested that among couples who do not share the same views on the continuation of the practice, the opinion of the mother is significant in determining the FGM status of a daughter. FGM is found to be more prevalent among daughters whose mothers want FGM to continue and fathers are opposed or undecided, compared to daughters whose fathers are the sole parent supporting its continuation.

While a daughter's likelihood of being cut is much higher when both parents think the practice should continue, the analysis also shows that many cut girls have parents who oppose the practice. This finding seems to confirm that parental opinions may not always regulate daughters' cutting, which could instead be determined by extended family or community members, as some existing research suggests [12]. In certain situations, the opinions of the parents might not have an impact upon if, how, when, and by whom a girl is cut, and a cultural shift at the community level would be necessary to eradicate the practice.

While the analysis provided insights on couples' opinions and the potential impact they have on decisions surrounding daughters' circumcision, additional country specific research is required to understand how power dynamics within a couple impact decisions on FGM. This would allow for more targeted programmatic interventions aimed at promoting the abandonment of the practice.

## Supporting information

**S1 Table. Percentage distribution of couples with at least one living daughter aged 0 to 14 years, by whether they have concordant or discordant opinions about the continuation of the practice.**
(DOCX)

**S2 Table. Percentage of daughters who have undergone FGM, by parental opinions about the continuation of the practice.**
(DOCX)

## Acknowledgments

The authors thank Ivana Bjelic and Munkhzul Zookhuu for their data processing support.

## Author Contributions

**Conceptualization:** Claudia Cappa.

**Data curation:** Claudia Cappa, Claire Thomson.

**Formal analysis:** Claudia Cappa, Claire Thomson.

**Methodology:** Claudia Cappa.

**Supervision:** Claudia Cappa.

**Writing – original draft:** Claudia Cappa, Claire Thomson.

**Writing – review & editing:** Claudia Cappa, Colleen Murray.

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
