## [Decision Letter · Decision Letter 0]

4 Feb 2020

PONE-D-19-35066

Understanding the association between parental attitudes and the practice of female genital mutilation among daughters

PLOS ONE

Dear Dr. Cappa,

Thank you for submitting your manuscript to PLOS ONE. It took some time for me to secure two reviews from qualified reviewers. Both reviewers think that your piece is interesting and should be publishable after some revisions. R1 recommends a more thorough literature review. I suggest you have a look at a PLOS ONE paper I edited last year, https://journals.plos.org/plosone/article/comments?id=10.1371/journal.pone.0213380

R1 also gives suggestions how to improve the discussion section. R2 's comments focus on the analysis and the presentation of results. I generally agree with both reviewers and invite you to submit a revised version of the manuscript that addresses the points raised during the review process.

We would appreciate receiving your revised manuscript by Mar 20 2020 11:59PM. To enhance the reproducibility of your results, we recommend that if applicable you deposit your laboratory protocols in protocols.io, where a protocol can be assigned its own identifier (DOI) such that it can be cited independently in the future. For instructions see: http://journals.plos.org/plosone/s/submission-guidelines#loc-laboratory-protocols

We look forward to receiving your revised manuscript.

Kind regards,

Kimmo Eriksson, Ph.D.

Academic Editor

PLOS ONE

Journal Requirements:

2. Please consider revising your Abstract and make sure it it structured in terms of clearly defined background, objective, methods, result, conclusions. Refer to our submission guidelines for further guidance (https://journals.plos.org/plosone/s/submission-guidelines#loc-abstract). In addition, please refrain from stating p values as 0.000, either report the exact value or use the format p<0.0001.

4. Please amend your manuscript to include your abstract after the title page.

Reviewers' comments:

Reviewer's Responses to Questions

**Comments to the Author**

1. Is the manuscript technically sound, and do the data support the conclusions?

Reviewer #1: Partly

Reviewer #2: Partly

2. Has the statistical analysis been performed appropriately and rigorously? 

Reviewer #1: Yes

Reviewer #2: Yes

3. Have the authors made all data underlying the findings in their manuscript fully available?

Reviewer #1: Yes

Reviewer #2: Yes

4. Is the manuscript presented in an intelligible fashion and written in standard English?

Reviewer #1: Yes

Reviewer #2: Yes

5. Review Comments to the Author

Reviewer #1: This is an interesting and valuable paper on analysis of DHS data from multiple countries on parental attitudes toward and support for FGM. While the analysis is sound, I found that both the introduction to the paper and the (lack of) discussion at the end limit its value to the field. Overall, I felt that the authors needed to provide a more detailed summary of recent published research on FGM, of which there is a considerable quantity including from the countries of interest in this study and including evidence of effective intervention strategies. Also, there is no discussion section. The authors state conclusions, but these are really just a restatement of the results section. There is no interpretation of findings, discussion of potential future research, or detail on future interventions or approaches that might be implied by this research. More detail needs to be provided on why this study matters to the field. The analysis is sound, but it's not clear why we should care. What does this study really tell us, and where does it suggest we should go from here?

Reviewer #2: This article analyses the relationship between FGM and parental attitudes towards FGM in African countries where FGM is more or less common. The most interesting, if somewhat depressing, result is that mother's attitude trumps fathers attitude when they disagree. I find the paper convincing, well thought through and well written, though with substantial shortcomings that need to be addressed before I can recommend an accept.

1. Type of FGM is not included in the analyses. This could make a huge difference as some earlier studies have revealed a higher commitment to more costly rituals. Conversely, it may be easier to maintain a less costly commitment. Either way, type of FGM should be included in the analyses.

2. These results can be much more transparently presented in graph format. It is unnecessarily difficult to go through all results in table form. Please rectify this.

3. Where are the results from the statistical tests? It is stated in the text that there should be tests included, but these results are nowhere to be found. Thus, I don't really know if the results are what the authors say they are or not.

These shortcomings are of a magnitude that I cannot recommend acceptance, unless a major review is carried out. But the results seem interesting, if they are as presented, so please go through the work to do this.

6. PLOS authors have the option to publish the peer review history of their article (what does this mean?). If published, this will include your full peer review and any attached files.

Reviewer #1: No

Reviewer #2: No

---

## [Author Response · Author response to Decision Letter 0]

19 Mar 2020

Reviewer #1

This is an interesting and valuable paper on analysis of DHS data from multiple countries on parental attitudes toward and support for FGM. While the analysis is sound, I found that both the introduction to the paper and the (lack of) discussion at the end limit its value to the field. Overall, I felt that the authors needed to provide a more detailed summary of recent published research on FGM, of which there is a considerable quantity including from the countries of interest in this study and including evidence of effective intervention strategies. 

The section with the literature review has been revised significantly to add more detailed summary of recently published research on FGM. 

Also, there is no discussion section. The authors state conclusions, but these are really just a restatement of the results section. There is no interpretation of findings, discussion of potential future research, or detail on future interventions or approaches that might be implied by this research. More detail needs to be provided on why this study matters to the field. The analysis is sound, but it's not clear why we should care. What does this study really tell us, and where does it suggest we should go from here?

A discussion section was added. 

Reviewer #2

This article analyses the relationship between FGM and parental attitudes towards FGM in African countries where FGM is more or less common. The most interesting, if somewhat depressing, result is that mother's attitude trumps fathers attitude when they disagree. I find the paper convincing, well thought through and well written, though with substantial shortcomings that need to be addressed before I can recommend an accept.

1. Type of FGM is not included in the analyses. This could make a huge difference as some earlier studies have revealed a higher commitment to more costly rituals. Conversely, it may be easier to maintain a less costly commitment. Either way, type of FGM should be included in the analyses.

We did look at data on the circumstances surrounding the practice, including girl’s age at cutting, practitioner and type of cutting. The original intent was to see whether such circumstances would show differentials by parental attitudes. The hypothesis there was that, for instance, among couples with discordant opinions, the cutting may still happen but in milder forms or at younger age or with the involvement of a health professional as opposed to a traditional circumciser. 

We later decided not to show the results in the paper for a number of reasons. Circumstances do not seem to change drastically by parental attitudes. Most daughters underwent FGM when they were 0 to 4 years old and in none of the countries did age at cutting significantly change by parents’ opinions. The majority of girls were cut by traditional practitioners regardless of parental opinions towards the practice. Finally, there was a certain level of variations across the countries in terms of types of FGM covered in the surveys. The only category that was constant was infibulation, which was rare in the countries with data. Finally, due to the low number of cut daughters in some countries, the analyses on age, practitioner and type of FGM by parental attitudes had a very large level of uncertainly and many values had to be suppressed, which made the results less reliable, and therefore not that meaningful.

2. These results can be much more transparently presented in graph format. It is unnecessarily difficult to go through all results in table form. Please rectify this.

Two graphs were added to illustrate the findings in tables 2 and 3. 

3. Where are the results from the statistical tests? It is stated in the text that there should be tests included, but these results are nowhere to be found. Thus, I don't really know if the results are what the authors say they are or not.

The statement in the article was rectified to reflect only the actual type of analyses that were performed. 

Additional requirements:

 Style requirements were reviewed and changed were made to reflect the requirements. 

2. Please consider revising your Abstract and make sure it it structured in terms of clearly defined background, objective, methods, result, conclusions. Refer to our submission guidelines for further guidance (https://journals.plos.org/plosone/s/submission-guidelines#loc-abstract). In addition, please refrain from stating p values as 0.000, either report the exact value or use the format p<0.0001.

 The abstract was revised to follow the suggested structure. P values were modified. 

An ORCID was created and validated. 

4. Please amend your manuscript to include your abstract after the title page.

The abstract was added after the title page.

---

## [Editor Report · Decision Letter 1]

20 Mar 2020

PONE-D-19-35066R1

Understanding the association between parental attitudes and the practice of female genital mutilation among daughters

PLOS ONE

Dear Dr. Cappa,

Thank you for submitting your revised manuscript to PLOS ONE. I have read it and I don't think I need to send it out on further review. However, I have some minor concerns I want you to address in a second revision.

1. In your response to Reviewer 2, you give reasons for not distinguishing between different types of cutting. I accept these reasons. However, in the method section of the manuscript you need to report what types of cutting are included in the FGM measure you report, and you need to briefly state why different types are treated as a single category in your analysis.

2. In the Figure 1 note, I think the sentence "Due to rounding, values will not add exactly to 100%" should be removed (it only applies to the table, right?). 

3. In the Figure 1 note, add a sentence about the accuracy of these percentages, e.g., "The width of 95% confidence intervals for these percentages were typically just a few percentage points, see Table S1."

4. In Figure 2, confidence intervals should be included as error bars in the bar diagram. 

5. State explicitly which null hypothesis the p-values refer to. My guess is that the null hypothesis (that you reject in all countries except one) is "the prevalence of FGM does not depend on parents' opinions". 

We would appreciate receiving your revised manuscript by May 04 2020 11:59PM. To enhance the reproducibility of your results, we recommend that if applicable you deposit your laboratory protocols in protocols.io, where a protocol can be assigned its own identifier (DOI) such that it can be cited independently in the future. For instructions see: http://journals.plos.org/plosone/s/submission-guidelines#loc-laboratory-protocols

We look forward to receiving your revised manuscript.

Kind regards,

Kimmo Eriksson, Ph.D.

Academic Editor

PLOS ONE

---

## [Author Response · Author response to Decision Letter 1]

1 May 2020

1. In your response to Reviewer 2, you give reasons for not distinguishing between different types of cutting. I accept these reasons. However, in the method section of the manuscript you need to report what types of cutting are included in the FGM measure you report, and you need to briefly state why different types are treated as a single category in your analysis.

A sentence was added to describe the type of FGM included in the analysis. 

2. In the Figure 1 note, I think the sentence "Due to rounding, values will not add exactly to 100%" should be removed (it only applies to the table, right?). 

The sentence above was removed. 

3. In the Figure 1 note, add a sentence about the accuracy of these percentages, e.g., "The width of 95% confidence intervals for these percentages were typically just a few percentage points, see Table S1."

Confidence intervals were added to Figure 1. 

4. In Figure 2, confidence intervals should be included as error bars in the bar diagram. 

Confidence intervals were added to Figure 2. 

5. State explicitly which null hypothesis the p-values refer to. My guess is that the null hypothesis (that you reject in all countries except one) is "the prevalence of FGM does not depend on parents' opinions". 

The null hypothesis was made clear in the section on methods and the main finding in terms of statistical association was emphasized in the section on results.

---

## [Editor Report · Decision Letter 2]

5 May 2020

Understanding the association between parental attitudes and the practice of female genital mutilation among daughters

PONE-D-19-35066R2

Dear Dr. Cappa,

We are pleased to inform you that your manuscript has been judged scientifically suitable for publication and will be formally accepted for publication once it complies with all outstanding technical requirements.

With kind regards,

Kimmo Eriksson, Ph.D.

Academic Editor

PLOS ONE
---

## [Editor Report · Acceptance letter]

12 May 2020

PONE-D-19-35066R2 

Understanding the association between parental attitudes and the practice of female genital mutilation among daughters 

Dear Dr. Cappa:

I am pleased to inform you that your manuscript has been deemed suitable for publication in PLOS ONE. Congratulations! Your manuscript is now with our production department. 

With kind regards,

on behalf of

Prof. Kimmo Eriksson 

Academic Editor

PLOS ONE